# THIS STATE LOOKS LIKE THAT: SELF-INTERPRETABLE REINFORCEMENT LEARNING AGENTS USING PROTOTYPE SOFT ACTOR-CRITIC

## ABSTRACT

Reinforcement learning (RL) has achieved remarkable success across complex decision-making tasks, especially with the advent of deep neural networks. However, the resulting models are often opaque, making their deployment in safety-critical domains challenging. Explainable AI aims to address this issue, but most specific efforts for deep RL remain limited either to post-hoc explanation methods or to imitation learning and distillation procedures. These latter approaches rely on pre-trained black-box agents and are typically restricted to environments with discrete action spaces, limiting their scalability and interpretability. In this paper, we introduce ProtoSAC, a novel deep RL architecture that integrates a prototype-based actor into the Soft Actor-Critic (SAC) algorithm, enabling intrinsic interpretability in continuous action spaces. Our method learns a set of prototypes that represent interpretable state clusters, each associated with a Gaussian action distribution. Actions are generated as a similarity-weighted mixture over these prototypes, providing transparent decision-making without sacrificing performance. We evaluate ProtoSAC on continuous control environments and show that it matches the performance of the original SAC while offering enhanced interpretability.

## 1 INTRODUCTION

Deep reinforcement learning (DRL) models reach remarkable capabilities in solving complex sequential decision-making tasks (Silver et al., 2016), achieving super-human performance in areas such as game playing, robotics (Kober et al., 2013), and autonomous control (Kiumarsi et al., 2017). The incorporation of deep neural networks into reinforcement learning (RL) frameworks expands their scope and efficacy, allowing them to handle high-dimensional input spaces. However, despite these significant advancements, DRL models suffer from a crucial limitation: their decision-making processes often remain opaque and unintelligible to human observers, preventing their adoption in safety-critical and trust-dependent applications.

To address this limitation, eXplainable Artificial Intelligence (XAI) emerges as a critical research area aiming to enhance the transparency and interpretability of otherwise black-box models. Within XAI, the subfield of explainable DRL specifically focuses on elucidating the decision-making processes of RL agents by either post-hoc interpretability techniques or designing self-interpretable agents from the ground up. However, the majority of existing explainable methodologies predominantly focus on classical RL problems rather than DRL settings, and often rely heavily on post-hoc explanations that attempt to interpret already trained, opaque models. While post-hoc methods can offer insights into agent behavior, they are generally less faithful, as they do not reflect the true internal decision-making process of the model but instead approximate it. In contrast, self-explainable agents are designed to make their reasoning inherently transparent, offering higher fidelity in the explanations and stronger guarantees of alignment between the model's behavior and the provided interpretations.

Few studies explore the concept of self-interpretable DRL agents. Among these, our work is closely related to the ones of Kenny et al. (2023) and Borzillo et al. (2023). They propose two architectures, namely PW-Net and Shared PW-Net, which employ prototypical classifiers to provide explanations

for RL agents through distillation and imitation learning. These approaches leverage prototype-based interpretability, a class of techniques that use prototypes (i.e., representative parts or full instances of the training data) to ground the inference process. However, both PW-Net and Shared PW-Net rely on the pre-training of a conventional black-box agent, which is subsequently distilled into an interpretable agent. This limits the learning capabilities to what the black-box model has already learned.

In this paper, we propose a novel self-interpretable DRL model designed specifically to address the limitations identified in prior works in continuous action-space scenarios. Our approach integrates a prototype-based component within the Soft Actor-Critic (SAC) algorithm (Haarnoja et al., 2018). Specifically, we design an actor network that integrates a prototypical layer to enable prototype-based decision-making. This layer directly compares input states against a set of learned prototypes and determines the action based on the resulting similarity scores. Each prototype corresponds to a specific action distribution characterized by mean and standard deviation parameters. The final action is produced as a similarity-weighted combination of these prototypes. By embedding interpretability directly within the policy learning framework, our approach provides intrinsic interpretability from scratch, avoiding the need for distillation of pre-trained agents, and fully supports continuous action spaces.

Overall, the main contributions of our work are as follows: 1) we propose *ProtoSAC*, the first self-interpretable reinforcement learning agent that integrates a prototype-based actor directly into the SAC framework for continuous action spaces; 2) we design a novel architecture where the actor outputs are defined as Gaussian distributions through a similarity-weighted combination over learned prototypes, providing transparent and case-based decision-making; 3) we introduce a prototype update mechanism and additional regularization losses that enhance coverage, diversity, and interpretability of the prototypes throughout training; 4) we conduct an experimental evaluation across benchmark environments to assess *ProtoSAC*'s performance and interpretability, including ablation studies on loss terms and prototype quantity; 5) we provide an open-source implementation of our proposed method and experimental setup[1].

The remainder of this work is structured as follows: we begin by reviewing the current literature on explainable artificial intelligence and interpretable reinforcement learning; we then formalize the theoretical background including SAC and prototype-based models; subsequently, we introduce the architecture and training procedure of *ProtoSAC*; we follow with an experimental evaluation on continuous action-space environments; finally, we conclude by summarizing our findings and discussing future research directions.

## 2 RELATED WORK

In this section, we review related state-of-the-art methodologies, first discussing the broader field of XAI and subsequently focusing on its application to DRL.

**Explainable Artificial Intelligence.** XAI is a research field focusing on making the decisions of AI models understandable to humans. XAI methods can be primarily categorized into two classes: post-hoc methods and self-explainable models (Došilović et al., 2018). Post-hoc methods aim at providing explanations for already trained black-box models, typically by analyzing model predictions after training has been completed. In contrast, self-explainable models focus on designing architectures that are interpretable by construction, embedding transparency into the model structure itself and thus enabling the generation of explanations simultaneously with predictions. Our work specifically targets the development of a self-explainable model within the domain of DRL.

In the literature of self-explainable approaches, two primary directions are explored: concept-based models (Koh et al., 2020; Wang et al., 2023a) and prototype-based models (Chen et al., 2019; Wang et al., 2023b). Concept-based models attempt to identify and leverage meaningful human-interpretable concepts present in the input data. The predictions of these models are subsequently derived based on the presence or activation of such concepts. However, a significant limitation of these methods is the necessity of pre-labeled datasets with concept annotations, a requirement that is challenging to fulfill in many scenarios. In contrast, prototype-based models do not require pre-labeled concepts, as they automatically learn exemplary instances, known as prototypes, directly

---

[1]Public GitHub repository: **URL scrubbed for double-blind reviewing. Code available in suppl. mat.**

Table 1: Comparison of prototype-based approaches for explainable DRL.

| Approach | Self-Interpretable | Learning | Prototype Selection |
|---|---|---|---|
| ProtoX | No | - | Automatic |
| PW-Net | Yes | IL | Manual |
| Shared-PW-Net | Yes | IL | Automatic |
| ProtoSAC (Ours) | Yes | RL | Automatic |

during the training phase. Predictions are then made based on the similarity between the input and these learned prototypes, which naturally provides intuitive explanations to human users.

**Explainable Deep Reinforcement Learning.** Within the context of DRL, explanations can be provided at different stages of the decision-making process. Some methods focus on clarifying the reasoning behind the agent's actions (actor explanations), others interpret environment dynamics (world explanations), and some approaches explicitly explain how the reward signals shape the agent's behavior (reward explanations) (Qing et al., 2023; Milani et al., 2024; Glanois et al., 2024). In particular, we focus on those approaches that aim at generating an interpretable actor, that is able to provide explanations together with decisions. Most of these approaches proceed by either distilling pre-trained models or through an imitation learning (IL) approach. Many approaches for instance, propose to approximate the policy and the Q-value functions using decision trees, programmatic policies or even logic (Bastani et al., 2018; Verma et al., 2018; Frosst & Hinton, 2018). However, these approaches increase the interpretability by approximating deep neural networks with simpler models. Delfosse et al. (2024), instead propose a concept-based approach that grounds the input state to human-designed concepts and learns logic rules over these concepts. Despite the fact that in this case the concepts are actually learned using deep neural networks, this work requires a manual labeling of the concepts, which is not actually feasible in all cases. A novel research direction involves prototype-based approaches. These methods originate from architectures initially developed for image classification, such as ProtoPNet (Chen et al., 2019), where a learnable weight matrix stores representative samples, called prototypes, from the training set. The model computes its output based on the similarity between the input and these prototypes. Ragodos et al. (2022) present one of the first applications of prototype-based methods in explainable DRL through ProtoX. However, their method is post-hoc: it explains a black-box agent rather than producing an inherently interpretable one. PWNet (Kenny et al., 2023) represents the first attempt to build self-explainable DRL agents using prototypes. Kenny et al. (2023) adopt an IL framework, transforming the reinforcement learning task into a supervised one, which enables the use of models like ProtoPNet. In PWNet, decisions are inferred by comparing the input state to a set of prototypes. A key limitation of this approach is its reliance on manually selected prototypes, which restricts its applicability in scenarios lacking prior domain knowledge. To address this, Borzillo et al. (2023) propose Shared-PWNet, which automates the prototype selection process. While promising, this method still depends on IL. In contrast, our approach, ProtoSAC, integrates the prototypical framework directly into the Soft Actor-Critic (SAC) algorithm. This allows us to constrain the agent's policy toward interpretable behaviors without requiring IL or manual prototype selection. To clarify the distinctions between these prototype-based approaches, Table 1 summarizes their main characteristics. Finally, it is worth mentioning that although prototypes have been explored in continuous action spaces (Yarats et al., 2021), they have not been explored for transparency. Unlike ProtoRL, which leverages them for exploration, our approach employs prototypes for case-based reasoning to enhance interpretability.

## 3 BACKGROUND

In this section, we define the mathematical basis for the understanding of our proposal. We start by introducing the reinforcement learning setting; we then follow with the description of the SAC algorithm; finally, we present the basis of prototype-based approaches.

**Reinforcement Learning** is a subfield of machine learning where an agent learns the correct behavior by interacting with the environment and leveraging reward signals to improve its decision-making over time. Formally, RL problems are often modeled as Markov Decision Processes (MDPs), defined by the tuple $(S, A, T, r, \gamma)$ where S is the set of states, A is the set of actions, $T(s', s, a)$ is the transition probability, $r(s, a)$ is the reward function, and $\gamma[0, 1)$ is the discount factor. At each time

step, the agent observes a state $s \in S$, selects an action $a \in A$ according to a policy $\pi(a|s)$, receives a scalar reward r(s,a), and transitions to the next state $s'$. The goal of the agent is to learn a policy that maximizes the expected discounted return $G_t$, defined as:

$$G_t = \mathbb{E}_\pi \left[ \sum_{t=0}^{\infty} \gamma^t r(s_t, a_t) \right] \qquad (1)$$

To achieve this objective, it is essential to estimate the long-term value of selecting specific actions in particular states. This is captured by the *Q-function* (or *action-value function*), which represents the expected return starting from a state $s$, taking an action $a$, and subsequently following a given policy $\pi$. Formally, the Q-function is defined as:

$$Q^\pi(s, a) = \mathbb{E}_\pi \left[ \sum_{t=0}^{\infty} \gamma^t r(s_t, a_t) \middle| s_0 = s, a_0 = a \right]. \qquad (2)$$

Modern RL algorithms often rely on function approximation (typically neural networks) to represent policies and value functions, enabling scalability to high-dimensional state and action spaces. Among these, *actor-critic methods* are particularly effective. In this setting, the *actor* selects actions based on a parameterized policy $\pi_\theta(a|s)$, while the *critic* estimates the value function, typically the action-value function $Q^\pi(s, a)$. In the remainder of this section, we go into the details of SAC, which is the algorithm on which we build upon to design our approach.

**Soft Actor-Critic** is an actor-critic algorithm designed for continuous action spaces (Haarnoja et al., 2018). It is based on the maximum entropy RL framework, which aims to not only maximize expected cumulative rewards but also encourages exploration by maximizing the entropy of the policy. The objective function in maximum entropy RL is defined as:

$$J(\pi) = \sum_{t=0}^{T} \mathbb{E}_{(s_t, a_t) \sim \rho_\pi} \left[ r(s_t, a_t) + \alpha \mathcal{H}(\pi(\cdot|s_t)) \right], \qquad (3)$$

where $\rho_\pi$ is the state-action distribution induced by the policy $\pi$, $\mathcal{H}(\pi(\cdot|s_t)) = -\mathbb{E}_{a_t \sim \pi}[\log \pi(a_t|s_t)]$ is the entropy of the policy at state $s_t$, and $\alpha$ is a temperature parameter that controls the trade-off between reward maximization and entropy. SAC maintains three types of neural networks: a stochastic policy network $\pi_\theta(a|s)$ (actor), two Q-value networks $Q_{\phi_1}(s, a)$ and $Q_{\phi_2}(s, a)$ (critics), and corresponding target networks for stability, which are exponential moving averages of the critics. The Q-functions are trained to minimize the soft Bellman residual. The target value is computed as:

$$y = r + \gamma \mathbb{E}_{a' \sim \pi(\cdot|s')} \left[ \min_{i=1,2} Q_{\bar{\phi}_i}(s', a') - \alpha \log \pi_\theta(a'|s') \right]. \qquad (4)$$

The loss for each critic is then:

$$\mathcal{L}_Q(\phi_i) = \mathbb{E}_{(s,a,r,s') \sim \mathcal{D}} \left[ (Q_{\phi_i}(s, a) - y)^2 \right], \quad i = 1, 2. \qquad (5)$$

The policy network is updated by minimizing the following loss:

$$\mathcal{L}_\pi(\theta) = \mathbb{E}_{s \sim \mathcal{D}, a \sim \pi_\theta} \left[ \alpha \log \pi_\theta(a|s) - \min_{i=1,2} Q_{\phi_i}(s, a) \right]. \qquad (6)$$

While the critic models are only used at training time, the policy model is the one that is actually used at prediction time to execute actions. For this reason, in this paper, we propose to substitute the policy model to make its decision process more transparent through the use of prototypes.

**Prototype-Based Methods** attempt to explain a model's decisions by measuring the similarity between a given input and a set of representative examples, known as prototypes. As a result, these methods are usually built by extending a pre-trained agent with an additional explainability network. This network typically defines $N$ distinct prototypes organized in a matrix $P \in \mathbb{R}^{N \times H}$, where $H_P$ is the prototype hidden dimension. The architecture is then composed of three interconnected components that transform raw input states into interpretable action predictions: an *encoder*, a *prototype layer*, and a final *prediction layer*. First, the *encoder* $f_{enc}(s)$ maps the observed state $s$ to a latent representation $z$:

$$z = f_{enc}(s). \qquad (7)$$

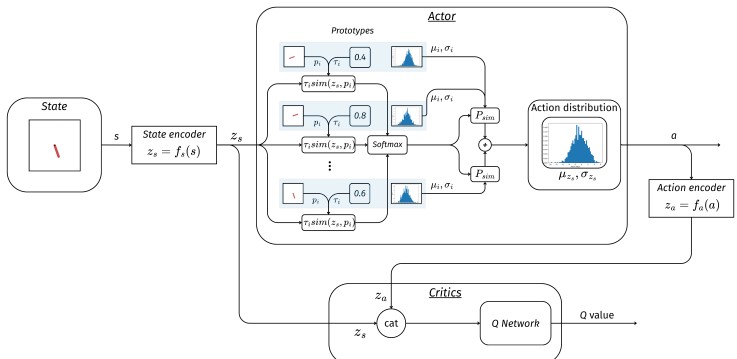

Figure 1: Overview of the *ProtoSAC* architecture. The state is encoded through a shared encoder into a latent representation, which is passed to the *ProtoActor* to compute similarities with learnable prototypes. The weighted combination defines a Gaussian distribution for action sampling. The *Critic* receives the encoded state and action to compute the Q-value.

The resulting representation is then compared against all the prototypes $p_i \in P$ of the *prototype layer*, thus computing a similarity score for each of them. This score measures how closely the projected latent representation aligns with the prototypes, providing a basis for interpretable decision-making:

$$\text{sim}(z, p_i) = \log \left( \frac{||z - p_i||_2 + 1}{||z - p_i||_2 + \epsilon} \right) \quad i \in [1, ..., N]. \tag{8}$$

Finally, these similarity scores are aggregated through a fully connected *prediction layer* to determine the final action, with weights $W \in \mathbb{R}^{N \times O}$, where O is the action dimension. This layer combines the prototype matching results to produce a set of action scores, from which the final action is selected:

$$a_j = \sum_{i=1}^{N} W_{i,j} sim(z, p_i) \quad j \in [1, ..., O]. \tag{9}$$

## 4    PROTOSAC

We introduce *ProtoSAC*, a modified version of the SAC algorithm that integrates a prototype-based decision mechanism directly into the actor network. This design enables explainable decision-making during both training and deployment. We start by illustrating the general architecture, and then we proceed by presenting the adopted training procedure. In Figure 1 we depict an overview of the architecture, and in the remainder of this section we go over the details of the various components.

**ProtoActor.** *ProtoSAC* retains the original SAC structure (consisting of an actor, two critics, and a replay buffer) but replaces the standard actor with a prototype-based actor, referred to as the ProtoActor. This actor is paired with a modified critic network, which is adapted to process the prototype-conditioned action representations while preserving the learning dynamics of the SAC algorithm. Inspired by existing prototype-based methods, the input state is first passed through a state encoder $f_s$ (shared by the actor and the critic), which maps the raw state into a latent representation $z_s$. This latent state is then fed into the ProtoActor. In order to produce continuous outputs, we pair the prototype matrix $P$ with three additional elements: an importance weight $\tau \in \mathbb{R}_{\geq 0}^{N}$, a mean $\mu \in \mathbb{R}^{N \times O}$, and a standard deviation $\sigma \in \mathbb{R}^{N \times O}$, which together parameterize a Gaussian distribution over actions for each of the prototypes. Given the encoded state $z_s$, the actor computes the similarity scores between the input and each prototype $sim(z_s, p_i)$ using Equation 8. Then, the similarity scores are used to produce a weighted combination of the prototype Gaussians by summing over the means $\mu_i$ and standard deviations $\sigma_i$, proportionally to both the similarity values and the importance weights. In this way, more similar and more important prototypes have a greater influence on the final action distribution. Formally, the final action distribution parameters are computed

as:

$$P_{\text{sim}_i} = \frac{\exp(\tau_i \cdot \text{sim}(z_s, p_i))}{\sum_{j=1}^{N} \exp(\tau_j \cdot \text{sim}(z_s, p_j))}, \quad i \in [1, ..., N] \tag{10}$$

$$\mu_{z_s} = \sum_{i \in N} P_{sim_i} \mu_i, \quad \sigma_{z_s} = \sum_{i \in N} P_{sim_i} \sigma_i. \tag{11}$$

$\mu_{z_s}$ and $\sigma_{z_s}$ define the parameters of the final Gaussian distribution from which the agent samples its action to be executed in the current state $s$ during interaction with the environment.

**Critic.** The sampled action from the ProtoActor is then processed by the action encoder $f_a$, which maps the continuous action into a latent representation. This step ensures a balanced representation between the encoded action and the encoded state. The critic, then, concatenates the encoded state $z_s$ and the encoded action $z_a$ with a multi-layer perceptron (MLP) to estimate the Q-value. The remaining training procedure follows the same steps described in the standard SAC framework.

**Training.** To allow the encoder to better capture the latent representation of the states, we initially train only the critic for a fixed number of steps before updating the actor (Korenkevych et al., 2024). This warm-up phase helps the encoder learn a meaningful representation of the state space, which in turn facilitates more stable policy learning once actor training begins. In this phase, the importance weights $\tau$ are updated to increase or decrease the influence of each prototype. Unlike standard SAC, *ProtoSAC* compares each encountered state to a fixed set of prototypes. As a result, the agent effectively interacts with a compressed view of the state space, which may limit the diversity of its experience. To mitigate this, *ProtoSAC* periodically updates its prototypes to better reflect underexplored regions of the environment. Every $M$ episodes, we identify the prototypes whose importance weight $\tau$ falls below the $q$ quantile across recent interactions (in our implementation, $q$ is set to 0.5 after empirical evaluation). These prototypes are considered under-performing or underutilized and are selected for replacement. New prototype candidates $p_{c_i}$ are sampled from the replay buffer by selecting states $s_i$ that are least similar to the current prototype set:

$$p_{c_i} = \underset{z_{s_i}}{argmax}(\frac{1}{\sqrt{\sum_{n \in N} sim(z_{s_i}, p_n) + \epsilon}}), z_{s_i} = f_s(s_i) \tag{12}$$

This encourages exploration of novel areas in the state space and helps avoid prototype redundancy. For each newly selected prototype, we compute the mean and standard deviation of the action distribution that would have been assigned had the state been encountered during normal training, following Equation 10. These parameters are then assigned to the prototype, along with a randomly initialized similarity weight $\tau$. The full prototype set is then updated accordingly. To improve prototype selection and ensure effective coverage of the state space, we introduce three supplementary loss terms to the actor's objective:

- a $\tau$ **Regularization Loss** ($\mathcal{L}_\tau$) encourages the model to use the least number of prototypes by penalizing the importance weights $\tau_i$. This ensures that only a small portion of prototypes are used, therefore leading to simple explanations:

$$\mathcal{L}_\tau = \sum_{i \in N} \tau_i; \tag{13}$$

- an **Orthogonality Loss** ($\mathcal{L}_{orth}$) enforces approximate orthogonality between prototypes. This ensures that they cover distinct directions in latent space rather than collapsing onto redundant regions:

$$\mathcal{L}_{\text{orth}} = \left\| PP^\top - I \right\|_F, \tag{14}$$

where $\| \cdot \|_F$ denotes the Frobenius norm and $I$ is the identity matrix.
- a **Negative Entropy Loss** ($\mathcal{L}_{ent}$) encourages each state to activate more than one prototype with significant similarity, avoiding overly "hard" assignments. If only a single prototype dominates, interpretability is weakened: the agent appears to always "copy" one prototype, rather than reasoning over multiple case-based examples:

$$\mathcal{L}_{ent} = \sum_{n \in N} P_{sim} log(P_{sim}), \tag{15}$$

where $P_{\text{sim}} \in \mathbb{R}^N$ denotes the softmax-normalized similarity distribution over prototypes for a given state, defined in Equation 10.

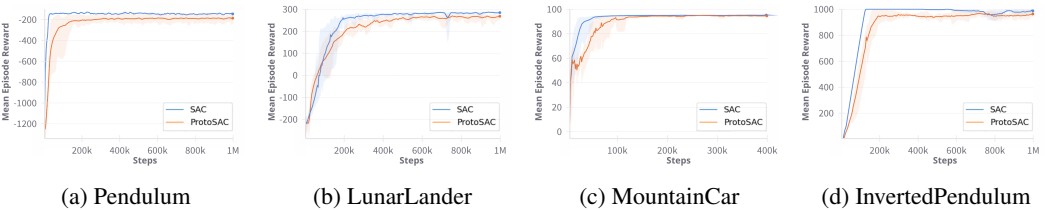

Figure 2: Performance comparison between *ProtoSAC* (orange) and SAC (blue), showing episode rewards over the course of training for each tested environment. Shaded areas represent the standard deviation across five runs.

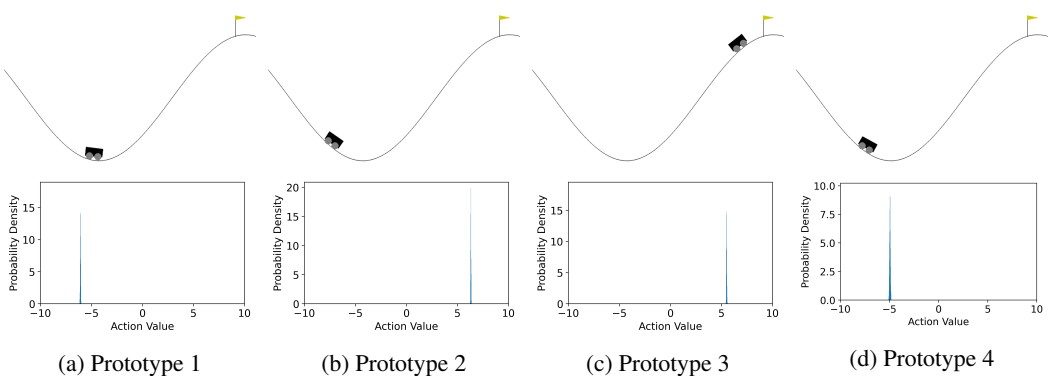

Figure 3: Examples of the 4 most important prototypes according to $\tau$ in the MountainCar environment. Each prototype is represented by its associated state and the action Gaussian distribution. A negative action is associated with a leftward motion (Figures a and d), while a positive action corresponds to a rightward movement (Figures b and c).

The final loss of *ProtoSAC* can be written as:

$$\mathcal{L} = \mathcal{L}_\pi + \alpha \mathcal{L}_{ent} + \gamma \mathcal{L}_{\text{orth}} + \gamma \mathcal{L}_\tau \tag{16}$$

where $\mathcal{L}_\pi$ is the original SAC actor loss (defined in Equation 6), $\alpha$ is the entropy coefficient, and $\gamma = 0.0001$.

## 5 EXPERIMENTS

In this section, we evaluate whether *ProtoSAC* can achieve performance comparable to the standard SAC algorithm. Additionally, we present and analyze key prototypes learned during training to better understand how the model represents different aspects of the task. Finally, we perform comparative experiments with a state-of-the-art self-explainable DRL method, Shared-PW-Net.

**Experiment Setup.** We test our model on 8 popular benchmark environments with increasing complexity: in , the agent's objective is to swing and balance a pendulum in the upright position by applying continuous torque; in MountainCarContinuous-v0, the agent must drive a car up a steep hill; in MuJoCo InvertedPendulum-v5, the objective is to balance a pole upright on a moving cart; finally, in LunarLanderContinuous-v3, the agent controls a lunar lander and must perform a soft landing on a target area; in MuJoCo HalfCheetah-v5, the agent controls a planar cheetah robot to run as fast as possible; in MuJoCo Hopper-v5, the agent must make a 2D one-legged robot hop forward without falling; in MuJoCo Humanoid-v5, the agent controls a high-dimensional biped to walk and stay upright; and in CarRacing-v3, the agent drives a car on a procedurally generated racetrack and must complete laps while staying on the track. In the first four environments the optimal behavior is well known, which facilitates clearer inspection and evaluation of the learned policies through our prototypical explanations, the others, represent instead complex tasks which are harder to learn. For the experiments, we use the Stable-Baselines3 library, and we implement

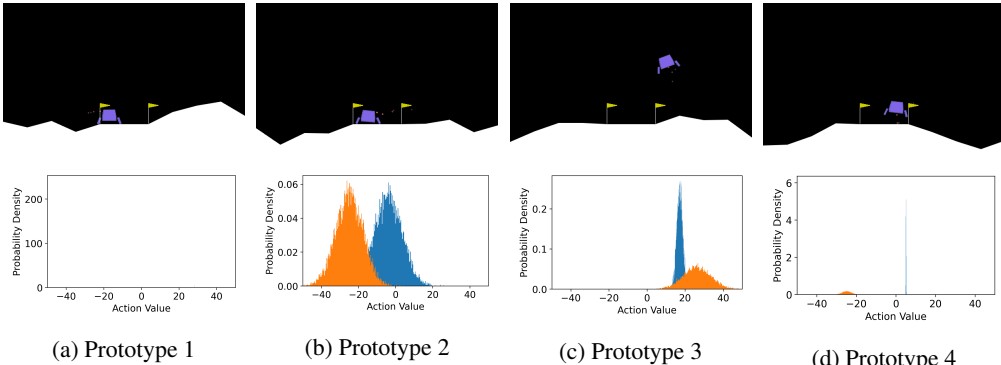

(a) Prototype 1      (b) Prototype 2      (c) Prototype 3      (d) Prototype 4

Figure 4: Examples of 4 most important prototypes according to $\tau$ in the LunarLander environment. Each prototype is illustrated with its corresponding state and the Gaussian actions distributions. The first action (blue) controls the main engine throttle, while the second (orange) controls the throttle of the lateral boosters.

our model to fit within it. Our hyperparameter selection is based on the recommended values available in the official GitHub repository[2]. Following the recommended configurations, in the case of the MountainCarContinuous-v0 environment, we integrate generalized State-Dependent Exploration (gSDE) (Raffin et al., 2022) in place of standard action noise to better handle exploration in continuous action spaces. We use a total of 30 prototypes for each model, except for the Inverted Pendulum, where we use 60 prototypes instead. The prototype set is updated every 20 episodes. Additionally, both the state and action encoders have a warm-up phase of 3000 steps. The full set of hyperparameters used for each environment and ablation studies over losses and prototypes are reported in the supplementary material.

**Training Results.** Our aim is to evaluate the impact of the prototype-based architecture on the SAC algorithm. Our approach cannot be directly applied to other actor–critic algorithms, as it relies on prototypes that represent action distributions, a mechanism specifically tied to SAC's stochastic policy formulation. Consequently, any meaningful comparison must be carried out against the SAC baseline. Figure 2 presents the training performance of *ProtoSAC* compared to the SAC baseline across the first four continuous control environments. Training performance is measured using the average cumulative reward evaluated over 5 independent runs. We observe that *ProtoSAC* achieves comparable performance to the baseline, although it generally requires more training steps to converge. While the peak performance reached by *ProtoSAC* is slightly lower in some environments, the gap is minimal. Given that *ProtoSAC* additionally provides interpretability throughout the learning process, this trade-off is arguably favorable. An additional observation is the increased variability in performance across training runs for *ProtoSAC* compared to the SAC baseline. This variability is more pronounced during the early stages of training but tends to diminish as the model stabilizes. This higher variance is likely due to the method's sensitivity to prototype initialization: when prototypes are not sufficiently representative, they can interfere with optimal learning and result in suboptimal performance. We observe that in simpler environments such as MountainCar, the performance gap between *ProtoSAC* and the baseline is negligible. However, as the environment becomes more complex, as in LunarLander, the gap increases slightly. Nevertheless, *ProtoSAC* still achieves performance that remains comparable to the baseline.

**Prototypes.** To understand how the agent's policy interacts with the environment and how the model captures the most relevant parts of the state space, we visualize the 4 prototypes with the highest $\tau$ scores. These represent the most influential components in the policy's decision-making process. Each prototype consists of a representative state (visualized as an image) and an associated action distribution, modeled as a Gaussian over the action space. Figure 3 shows the most important prototypes according to $\tau$ for the MountainCarContinuous environment. In this task, the agent must build momentum through oscillating in order to reach the hilltop. Several prototypes, such as Figures 3a and 3d, are associated with leftward motion (i.e., negative actions), aimed at gathering momentum.

---

[2]https://github.com/DLR-RM/rl-baselines3-zoo/blob/master/hyperparams/sac.yml

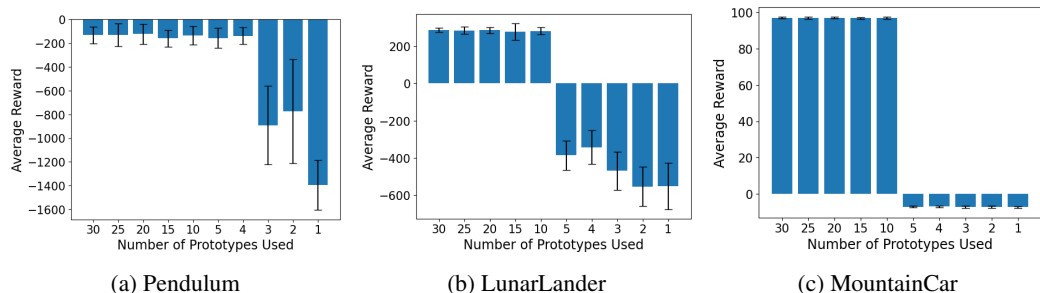

|  | (a) Pendulum | (b) LunarLander | (c) MountainCar |

Figure 5: Sensitivity analysis with different number of prototypes. We remove from each pretrained model the least used prototypes according to $\tau$ and evaluate the resulting performance.

Table 2: Comparison of prototype-based approaches for explainable DRL. We report the mean and standard deviation of rewards over 30 simulations. Best values are highlighted in bold, while values within one standard deviation of the best values are underlined.

| Environment | SAC | Shared-PW-Net | ProtoSAC |
|---|---|---|---|
| Pendulum | **-141.79 ± 61.79** | -524.33 ± 386.99 | -152.24±83.50 |
| Lunar Lander | 256.59±88.26 | 258.77±55.17 | **282.42±19.90** |
| Mountain Car | 97.55±0.27 | **97.63±0.30** | 95.29±0.77 |
| Inverted Pendulum | **1000.0 ± 0.0** | 33.43±10.37 | **1000.0 ± 0.0** |
| Hopper | $3380.251 \pm 1.980$ | $3374.412 \pm 3.049$ | **$3386.222 \pm 8.855$** |
| HalfCheetah | **$11098.333 \pm 150.083$** | $10369.168 \pm 1248.026$ | $9875.183 \pm 77.872$ |
| Humanoid | $4574.069 \pm 1358.091$ | $496.586 \pm 133.035$ | **$4953.814 \pm 3.268$** |
| CarRacing | $232.968 \pm 143.422$ | $-32.528 \pm 9.124$ | **$369.047 \pm 189.355$** |

Others, including Figures 3b and 3c, correspond to rightward thrusts (i.e., positive actions) intended to exploit the momentum and reach the goal. Figure 4 presents four of the most important prototypes for LunarLanderContinuous environment. Here, each action is a two-dimensional vector composed of the main engine throttle and lateral booster control. Consequently, the action distribution for each prototype is a two-dimensional Gaussian. The visualized prototypes illustrate how the agent has learned to perform fine-grained control in a highly dynamic setting. For example, Figures 4c and 4d show how the model balances the main engine and lateral boosters to align with the target. In contrast, Figures 4a and 4b illustrate two examples of landings: the first corresponds to a state in which the agent remains stationary, suggesting it has learned to recognize when inaction is optimal; the second shows a landing that requires balancing between the boosters and the main engine to avoid crashing. Here it is important to highlight that while the shuttle position is similar on the map, the difference lies in its current speed, indicated by the red sparks, and also inspectable quantitatively when running the environment. Overall, these examples highlight the model's ability to represent diverse behaviors and adapt its policy to the complexity of the task.

**Sensitivity Analysis.** Here, we assess the model's capacity in retaining only meaningful prototypes, in order to ensure sparse and simple explanations. We therefore study impact of each prototype on performance by progressively removing the least-used prototypes according to the importance weight $\tau$. In Figure 5, we report the results of this analysis for the Pendulum, LunarLander and MountainCar environments. In all the three environments reward remains stable even when several prototypes are removed, only declining noticeably after a considerable number are eliminated. In particular, in the simple task of Pendulum, 5 prototypes are sufficient to retain the performances of the full model, whereas in more complex tasks such as LunarLander, the amount of needed performances raises to 10. This observation suggests that the agent is capable of using $\tau$ to select a small subset of most important prototypes. This makes it sufficient to only inspect the prototypes with highest associated $\tau$ values to capture and explain the behavior of the agent, and it is not necessary to examine all prototypes to understand its decision-making.

**Comparison with Shared-PW-Net.** We now compare the performances of our model with those of another self-explainable model, Shared-PW-Net. As shown in Table 1, Shared-PW-Net is the only existing prototype-based approach that allows training self-explainable agents in continuous action spaces without requiring manual prototype selection. In Table 2, we report the cumulative rewards

Table 3: Explanation fidelity evaluation. Mean squared deviation ($\pm$ std) between original action and action after removing the most active prototype.

| Environment | Shared-PW-Net | ProtoSAC |
|---|---|---|
| Hopper | $0.182 \pm 0.174$ | $\mathbf{0.534 \pm 0.354}$ |
| HalfCheetah | $0.402 \pm 0.282$ | $\mathbf{0.696 \pm 0.528}$ |
| Humanoid | $0.050 \pm 0.044$ | $\mathbf{0.084 \pm 0.021}$ |
| CarRacing | $0.188 \pm 0.064$ | $\mathbf{0.383 \pm 0.525}$ |

for *ProtoSAC*, Shared-PW-Net, and SAC, evaluated on 30 simulations. The results show that *ProtoSAC* demonstrates consistently strong performance across all the environments. In the Pendulum environment, *ProtoSAC* outperforms Shared-PW-Net by a significant margin and performs competitively with SAC, even though with a higher variance. For Lunar Lander, *ProtoSAC* achieves the highest performance with the lowest variance. In the Mountain Car environment, while Shared-PW-Net attains the best score, the performance of all methods is near-optimal and the differences are marginal. Notably, in the Inverted Pendulum environment, *ProtoSAC* matches SAC's perfect score, while Shared-PW-Net fails to achieve meaningful performance, suggesting that *ProtoSAC* is better suited to high-reward precision tasks. This is mainly due to the fact that while the training of Shared-PW-Net is independent from the task, as it is only trained to replicate the behavior of the black-box, *ProtoSAC* is instead directly trained to solve the specific task, thus providing stronger performances in control tasks such as Pendulum and Inverted Pendulum. In high-dimensional continuous control environments, the results further confirm the robustness of *ProtoSAC*. In the Hopper, Humanoid and CarRacing tasks, *ProtoSAC* achieves the highest cumulative reward and exhibits low variance, indicating stable and effective control. On HalfCheetah, instead, although SAC remains the best-performing model, *ProtoSAC* still maintains competitive results compared to Shared-PW-Net, demonstrating its capacity to generalize to fast locomotion tasks.

**Exlanation Evaluation.** We evaluate the faithfulness of the explanations by measuring, for each model, the mean squared deviation between the original action and the action recomputed after removing the most active prototype. The results, reported in Table 3, show that *ProtoSAC* consistently exhibits larger deviations than Shared-PW-Net across all tasks, indicating a stronger functional dependency of the policy on the most active prototype. In Hopper and HalfCheetah, the deviation almost doubles for *ProtoSAC*, suggesting that its prototypes encode more decisive control patterns compared to those of Shared-PW-Net. A similar trend is observed in Humanoid and CarRacing, where *ProtoSAC* again yields higher deviations, which implies that its explanations are more tightly coupled to the underlying policy behavior.

## 6 CONCLUSIONS

In this work, we introduced *ProtoSAC*, a novel self-interpretable reinforcement learning architecture that integrates prototype-based reasoning directly into the SAC framework. Unlike previous approaches that rely on post-hoc explanations or imitation learning, ProtoSAC generates interpretable policies from scratch. By representing policies as similarity-weighted combinations over learned prototypes, our method enables transparent and intuitive decision-making. Through experiments on continuous action-space environments, we demonstrated that *ProtoSAC* achieves competitive performance compared to SAC and state-of-the-art self-explainable models, while offering explanations. Despite these promising results, our work has some limitations. *ProtoSAC* introduces additional complexity to the actor network, leading to slower convergence and higher variance in performance, especially in the early training phases. Furthermore, the quality of interpretability is strongly tied to the selection and diversity of the learned prototypes; poor initialization or suboptimal updates may hinder both learning and explainability. While we mitigate these issues with prototype replacement strategies and regularization losses, the training process remains more resource-intensive than standard SAC. Future research can address these limitations by investigating more efficient ways to initialize, update, and regularize prototypes, possibly using curriculum learning or meta-learning approaches. Moreover, combining prototype-based interpretability with other mechanisms, such as symbolic reasoning or concept bottlenecks, could yield richer explanations and greater robustness.

# 7 REPRODUCIBILITY STATEMENT

All the code needed to run and reproduce the experiments is available in the supplementary material attached to this paper. After acceptance, the material will be published on a public repository.

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

## A    TRAINING ALGORITHM

---

**Algorithm 1** Proto-Actor action selection

---

Input: initial policy parameters $\theta$, Proto-network parameters $p_i$ and their relative mean $\mu_i$ and standard deviation $\sigma_i$, the state encoder $f_s$
**repeat**
  Observe state $s$ and pass it through the state encoder $z_s \leftarrow f_s(s)$
  Confront the state with each prototype inside the Proto-Network :

$$\text{sim}(z_s, p_i) = \log\left(\frac{\|z_s - p_i\|_2 + 1}{\|z_s - p_i\|_2 + \epsilon}\right)$$

  Calculate the new action distribution

$$P_{\text{sim}_i} = \frac{\exp(\tau_i \cdot \text{sim}(z_s, p_i))}{\sum_{j=1}^{N} \exp(\tau_j \cdot \text{sim}(z_s, p_j))}, \quad i \in [1, ..., N]$$

$$\mu_z = \sum_{i \in N} P_{sim_i}\mu_i, \quad \sigma_z = \sum_{i \in N} P_{sim_i}\sigma_i$$

$$\theta \leftarrow \mathcal{N}(\mu_z, \sigma_z)$$

  Select the associated action a $\sim \pi_\theta(\cdot|s)$
**until** convergence

---

## B    EXPERIMENTAL SETTINGS AND REPRODUCIBILITY

In Table 4 we detail the hyperparameters used in the experiments. We perform our experiments on a machine equipped with a Intel(R) Xeon(R) CPU E5-2698 v4 @ 2.20GHz and a Tesla V100-SXM2-32GB. In Table 5 we report the average training times (in seconds) for *ProtoSAC* and SAC.

Table 4: Training hyperparameters for each environment.

| Hyperparameter | MountainCarContinuous-v0 | Pendulum-v1 | InvertedPendulum-v5 | LunarLanderContinuous-v3 |
|---|---|---|---|---|
| Timesteps | 400,000 | 1,000,000 | 1,000,000 | 1,000,000 |
| Learning Rate | 3e-4 | 1e-3 | 3e-4 | 7.3e-4 |
| Buffer Size | 50,000 | 1,000,000 | 1,000,000 | 1,000,000 |
| Batch Size | 512 | 256 | 256 | 256 |
| Entropy Coefficient | 0.1 | auto | auto | auto |
| Train Frequency | 32 | 1 | 1 | 1 |
| Gradient Steps | 32 | 1 | 1 | 1 |
| Gamma | 0.9999 | 0.99 | 0.99 | 0.99 |
| Tau | 0.01 | 0.005 | 0.005 | 0.01 |
| Learning Starts | 0 | 200 | 10,000 | 10,000 |
| Use SDE | True | - | - | |

Table 5: Average training times (in minutes) for SAC and ProtoSAC across the environments.

| Environment | SAC (min) | ProtoSAC (min) |
|---|---|---|
| Pendulum-v1 | 314 | 490 |
| LunarLanderContinuous-v3 | 378 | 540 |
| MountainCarContinuous-v0 | 157 | 204 |
| InvertedPendulum-v5 | 314 | 426 |

## C ABLATION STUDY

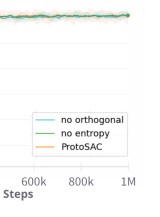 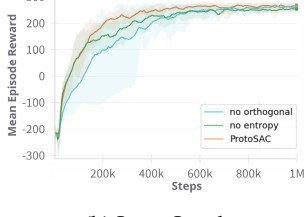 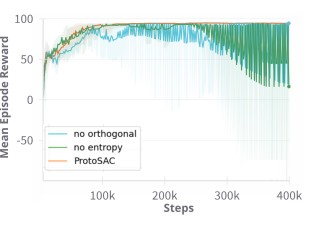

(a) Pendulum  (b) LunarLander  (c) MountainCar

Figure 6: The ablation study in the continuous environment. We can see that except for the Pendulum task, removing either the orthogonal loss (blue) or the entropy loss (green) tends to introduce high instability during training, although in some cases the final performance remains competitive.

To assess the contribution of each component of our method, we conduct an ablation study by systematically removing individual loss terms from the training process. Specifically, we evaluate the impact of the orthogonal loss and the negative entropy loss on the training dynamics across all continuous control environments. As shown in Fig. 6, these losses play a crucial role in maintaining training stability, particularly in more complex environments. While in the Pendulum environment the differences are marginal, starting from LunarLander, we observe greater variability in the learning curves when either loss is removed.

In particular, removing the orthogonal loss leads to highly unstable behavior—sometimes achieving high rewards, but often resulting in poor performance. For MountainCar, the most noticeable effect is not on peak reward, but on how long the model can train before suffering from overfitting. Since we employ generalized State-Dependent Exploration (gSDE), which adjusts the policy's standard deviation dynamically, excessive training can lead to saturation. In this setting, models trained with both auxiliary losses tend to delay overfitting, while those missing one of the losses experience instability sooner.

These findings suggest that the orthogonal loss and the negative entropy loss work in a complementary way: the orthogonal loss promotes diversity among prototypes, ensuring better coverage of the state space, while the negative entropy loss encourages the model to rely on more than one prototype with high similarity. Together, they help achieve more robust and generalized policies.

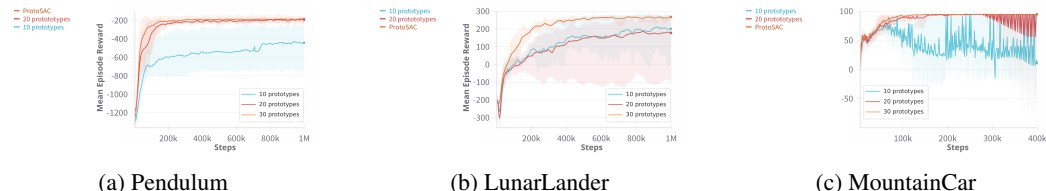

(a) Pendulum        (b) LunarLander        (c) MountainCar

Figure 7: Evaluate the impact of prototype quantity on performance. We compare ProtoSAC using three different numbers of prototypes: 30 (orange), 20 (red), and 10 (blue), across the three continuous control environments.

To determine the appropriate number of prototypes for each environment, we conducted an ablation study by training *ProtoSAC* with three different prototype counts: 30, 20, and 10. The goal was to assess how the number of prototypes influences overall performance and training stability.

In the simplest environment, Pendulum, using only 10 prototypes is sufficient to solve the task, although the performance does not fully match the baseline in terms of average reward. With 20 prototypes, the performance slightly decreases compared to 30, likely due to the limited complexity of the environment, where fewer prototypes can still provide adequate coverage of the state space.

In contrast, in the more complex LunarLander environment, reducing the number of prototypes to 20 or 10 results in significantly less stable training and often leads to poor task completion. A similar trend is observed in MountainCar, here using 20 prototypes allows the agent to solve the task, but with increased variance and reduced robustness compared to the 30-prototype setup, while using 10 prototypes fails to provide sufficient coverage, resulting in unsuccessful task completion.

## D  PENDULUM PROTOTYPES

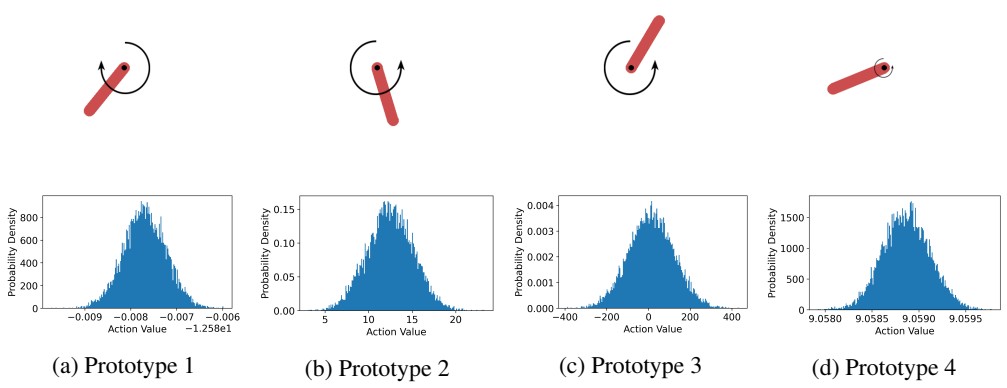

(a) Prototype 1   (b) Prototype 2   (c) Prototype 3   (d) Prototype 4

Figure 8: Examples of 4 most important prototypes according to $\tau$ in the Pendulum environment. Each prototype is represented by its associ-ated state and the action Gaussian distribution.

In this section, we analyze the prototypes used in the Pendulum environment, as shown in Figure 8. Each prototype is linked to a specific state and an associated action, corresponding to the torque applied to the pendulum. Interestingly, we observe a wide range of variances across prototypes. This variation can be interpreted as the model's way of encoding different levels of flexibility in decision-making. Prototypes with low variance suggest that, in those regions of the state space, the agent has learned a clear and consistent action pattern. In contrast, prototypes with high variance indicate regions where multiple actions may be reasonable, or where the agent has encountered more diverse experiences during training.

# E    LLMS STATEMENT

LLMs were employed for grammar checking and text polishing. Importantly, LLMs were not used to generate ideas for contributions or to retrieve literature.

