# OpenReview forum: "This State Looks Like That: Self-Interpretable Reinforcement Learning Agents using Prototype Soft Actor-Critic"
_ICLR.cc/2026/Conference — Submitted to ICLR 2026_

### Official Review · Reviewer_pKoK · 2025-10-22

**Soundness:** 2
**Presentation:** 2
**Contribution:** 2
**Rating:** 4
**Confidence:** 3

**Summary:**

This paper introduces ProtoSAC, a novel deep reinforcement learning architecture for continuous control tasks that is intrinsically interpretable. It integrates a prototype-based actor into the Soft Actor-Critic (SAC) framework. The agent learns a set of representative "prototype" states, each associated with an action distribution. Actions are then generated as a similarity-weighted mixture of these prototype policies, making the decision-making process transparent. The authors demonstrate that ProtoSAC matches the performance of the original SAC on several benchmark environments while providing clear explainability, a significant step beyond post-hoc methods or approaches limited to discrete action spaces.

**Strengths:**

1. Impressively matches the performance of a strong baseline (SAC) without a significant trade-off, directly addressing a key challenge in XAI.
2.   The "this state looks like that" reasoning framework is highly intuitive. Visualized prototypes and their weights offer clear, actionable insights into the agent's policy.

**Weaknesses:**

1.  The method introduces computational overhead compared to standard SAC due to similarity calculations, extra loss terms, and prototype management. The paper could benefit from a more detailed analysis of this overhead and its scalability.
2.  The model introduces several new hyperparameters (e.g., number of prototypes `K`, update frequency `M`, regularization coefficients `γ`). A sensitivity analysis or ablation study on these would strengthen the paper's claims and improve reproducibility.
3.  The authors note high variance in early training, which may be tied to prototype initialization. The work could be improved by exploring more robust initialization strategies to increase stability.

**Questions:**

1.  Have you investigated the trade-off between the number of prototypes `K`, model performance, and the granularity of the explanations? Could `K` be adapted dynamically based on environment complexity?
2. The current "hard replacement" strategy for updating prototypes might risk forgetting rare but critical edge cases. Have you considered "soft" update mechanisms, such as slowly updating a prototype's embedding with new, similar state features?
3. What is your perspective on scaling ProtoSAC to high-dimensional state spaces, such as image-based inputs (e.g., Atari)? What would be the main challenges for the encoder design and the similarity metric in such scenarios?

---

> ### Author Response · Authors · 2025-12-02
>
> - **The model introduces several new hyperparameters (e.g., number of prototypes K, update frequency M, regularization coefficients γ). A sensitivity analysis or ablation study on these would strengthen the paper's claims and improve reproducibility.**
>
> We ensure reproducibility by reporting all the parameters in the appendix. However, we would like to specify that although we empirically adapted some of them, most of the hyperparameters are fixed for all the environments.
>
> - **The authors note high variance in early training, which may be tied to prototype initialization. The work could be improved by exploring more robust initialization strategies to increase stability.**
>
> We thank the reviewer for the comment. Indeed, the variance in early training might be tied to prototype intialization. However we also observe that it generally converges after the first episodes, also thank to prototype update. Studying different options for prototype intialization might represent promising avenues for future studies.
>
> - **Have you investigated the trade-off between the number of prototypes K, model performance, and the granularity of the explanations? Could K be adapted dynamically based on environment complexity?**
>
> Also in response to other reviewers, we have included a study where we show that the model is completely capable of selecting the smallest amount of prototypes needed to solve the task. In the experiment, we show that turning off prototypes with low $\tau$ values does not affect performances: in some environments, performances are retained even when only leaving 5 prototypes active.
>
> - **The current "hard replacement" strategy for updating prototypes might risk forgetting rare but critical edge cases. Have you considered "soft" update mechanisms, such as slowly updating a prototype's embedding with new, similar state features?**
>
> The replacement only affects prototypes with low $\tau$ values that are not used. This can be seen as a "soft" update, as it does not completely erase previous prototypes. Future studies could investigate how this could affect forgetting in continuous learning scenarions, however, from our experiments we observe that prototype update is beneficial for the model.
>
> - **What is your perspective on scaling ProtoSAC to high-dimensional state spaces, such as image-based inputs (e.g., Atari)? What would be the main challenges for the encoder design and the similarity metric in such scenarios?**
>
> We would like to emphasize that ProtoSAC can take all types of input including 1-d vectors, images, or even sequences and graphs. Indeed prototype-based models were first developed with images and the image application would be straightforward. This is due to the fact that the state/prototype representation is handled by the state encoder $f_s$ which can be implemented as any neural network (MLP, CNN, ...).
> We evaluate this hypothesis in the CarRacing-v3 environment by training our model with a CNN encoder. We compare the performance with the baseline trained using the same hyperparameters, as well as with the Shared-PW-Net trained using the SAC baseline. As shown in the table below, ProtoSAC clearly outperforms Shared-PW-Net, which is unable to solve the environment.
>
>
>   | Environment       | SAC                   | Shared-PW-Net         | ProtoSAC              |
> |-------------------|------------------------|------------------------|------------------------|
> | CarRacing-v3        |  232.968 ± 143.422   |  -32.528 ± 9.124  |    **369.047 ± 189.355**   |
>
> - **The method introduces computational overhead compared to standard SAC due to similarity calculations, extra loss terms, and prototype management. The paper could benefit from a more detailed analysis of this overhead and its scalability.**
>
> The reviewer is correct, training times are slightly higher due to the constraints that induce interpretability. We already reported training times in the appendix of the original paper confirming this fact. However we would like to emphasize that this computational cost has to be paid in order to allow for having a model that is more interpretable. Additionally, while in distillation approaches the cost covers training the RL model and then distilling it, in our case we would only need to train the RL model as we do not necessitate distillation.

---

### Official Review · Reviewer_V6re · 2025-10-30

**Soundness:** 3
**Presentation:** 2
**Contribution:** 2
**Rating:** 2
**Confidence:** 4

**Summary:**

The paper introduces ProtoSAC, a variant of Soft Actor-Critic which integrates a prototype framework during training time rather than posthoc. ProtoSAC enables yielding intrinsic, case-based explanations by factorizing the action space as a similarity-weighted mixture of per-prototype Gaussians. Experiments on four continuous action space environments shows the method largely matches SAC performance while providing the above mentioned interpretability benefits.

**Strengths:**

* Prototype based SAC is both intuitive and enables drop in modification.
* For the environments tested, performance appears to hold versus the black box, with performance generally higher then the Shared-PW-Net baseline.
* The paper is generally well written and easy to understand.

**Weaknesses:**

* **W1** The novelty of the word is quite limited, appearing to be a straight forward application of prototype based frameworks to SAC. As the authors acknowledge, their work is very related work PW-Net and Shared PW-Net, the functional difference rl training versus distillation / imitation learning.
* **W2** The authors argue that distilling from a black box "limits the learning capabilities to what the black-box model has already learned," which is true, but no experiments in this paper convincingly show that directly applying PW nets during RL training is more scalable then distillation.
* **W3** The experimental results in the paper are extremely limited. Pendulum, Lunar Lander, Mountain Car, and Inverted Pendulum are broadly considered toy environments. This directly ties into W2: if the novelty of the approach rests on reinforcement learning being more scalable than imitation learning /distillation for learning prototype nets, experiments on more complicated environments should validate this. I would expect at the minimum experiments on MuJoCo.

* **W4** The work would benefit from a user study considering the primary objective of introducing interpretability to SAC. Anecodes in the form of Figures 3/4 are helpful but not enough to validate the utility of the learned prototypes.

* **W5** The ablation study as shown in Figure 5 is not convincing. Removing orthogonal loss (blue) or the entropy loss (green) does not seem to have an impact on final performance. The claim that, "These findings suggest that the orthogonal loss and the negative entropy loss work in a complementary way: the orthogonal loss promotes diversity among prototypes, ensuring better coverage of the
state space, while the negative entropy loss encourages the model to rely on more than one prototype
with high similarity. Together, they help achieve more robust and generalized policies." Its unclear how Figure 5 shows either of these claims.

**Questions:**

1. W2/3 would be cleared if the authors are able to demonstrate the method outperforms distillation / imitation learning on more complex environments, such as MuJoCo HalfCheetah-v4 or Procgen CoinRun.
2. Can the authors clarify what evidence supports their claims on line 645? It is unclear to me how variance in return during training implies better diversey among prototypes and better state space coverage.
3. How does the perceived interpretability of ablation variants compare to the proposed method? A user study here appears to be necessary.

---

> ### Author Response · Authors · 2025-12-02
>
> - **Demonstrate the method outperforms distillation / imitation learning on more complex environments**:
>
> We thank the reviewer for the suggestion. We included the following 4 additional complex environments: HalfCheetah, Humanoid, Hopper and CarRacing. We observe that even in this case our approach results in comparable or better performances wrt the black-box baseline and SharedPWNet.
>
> | Environment       | SAC                   | Shared-PW-Net         | ProtoSAC              |
> |-------------------|------------------------|------------------------|------------------------|
> | Hopper-v5          |  3380.251 ± 1.980   |  3374.412 ± 3.049  |    **3386.222 ± 8.855**   |
> | HalfCheetah-v5    |    **11098.333 ± 150.083**     |    10369.168 ± 1248.026      |  9875.183 ±  77.872   |
> | Humanoid-v5      |    4574.069 ± 1358.091    |  496.586 ± 133.035      |     **4953.814 ± 3.268**   |
> | CarRacing-v3        |  232.968 ± 143.422   |  -32.528 ± 9.124  |    **369.047 ± 189.355**   |
>
> - **These findings suggest that the orthogonal loss and the negative entropy loss work in a complementary way**
>
> As the orthogonal loss promotes diversity among prototypes, it ensuring better coverage of the state space, at the same time it would mean having only few prototypes active at the same time. The negative entropy loss balances this by encouraging the model to rely on more than one prototype with high similarity.
>
> - **User perception of the explanations.**
>
>     In this paper, we do not focus on a user's perception of interpretability with respect to the agent. Instead, our emphasis is on developing mathematical constraints within the model to enable inspection of its decision-making process. While there is active debate in the literature regarding how users perceive explanations, we deliberately avoid this analysis, because user-based evaluation often reflects expectations rather than uncovering true factors influencing the model. Nevertheless, in response to reviewers' suggestions, we provide a quantitative analysis of explanation fidelity, demonstrating how key prototypes affect the agent’s decisions. This approach confirms that the prototypes offer faithful explanations aligned with the actual behavior of the model. Specifically, we compare the fidelity of our explanations to those produced by Shared-PWNet in the following table.
>
>
> | Environment      | Shared-PW-Net        | ProtoSAC              |
> |------------------|-----------------------|------------------------|
> | Hopper-v5        | 0.182 ± 0.174         | **0.534 ± 0.354**      |
> | HalfCheetah-v5   | 0.402 ± 0.282         | **0.696 ± 0.528**      |
> | Humanoid-v5      | 0.050 ± 0.044         | **0.084 ± 0.021**      |
> | CarRacing-v3     | 0.188 ± 0.064         | **0.383 ± 0.525**      |
>
>
>    Additionally, regarding the question, "How easy would it be for a user to gain some kind of understanding of the model globally?", this reduces to identifying the most important prototypes and their associated Gaussian action distributions. Together, these elements provide a complete description of the model and enable a comprehensive understanding of its global behavior. More importantly, we observe, that thanks to our regularization losses, we force the model to use the smallest possible amount of prototypes, thus making explanations as simple as possible. We added in the paper where we show that the model retains strong performances even when least important prototypes (those associated with a small $\tau$) are removed.

---

### Official Review · Reviewer_i8Go · 2025-10-31

**Soundness:** 2
**Presentation:** 2
**Contribution:** 2
**Rating:** 4
**Confidence:** 4

**Summary:**

This paper introduces ProtoSAC, a novel self-interpretable reinforcement learning framework that integrates a prototype-based interpretation mechanism directly into the Soft Actor-Critic (SAC) framework for continuous control tasks. Its primary contribution is an intrinsically interpretable architecture where the actor's policy is defined by a similarity-weighted combination of learned prototypes, each representing an interpretable state cluster with an associated Gaussian action distribution. This design provides transparent, case-based decision-making without relying on post-hoc explanations or imitation learning from a black-box agent. The authors demonstrate that ProtoSAC achieves performance competitive with the standard SAC baseline and outperforms existing self-explainable methods like Shared-PW-Net across several benchmark environments, while also offering a prototype update mechanism and regularization losses to enhance interpretability and stability during training.

**Strengths:**

1. This paper presents an application of a prototype-based explanation framework (referred to as "ProtoSAC") to deep Reinforcement Learning, specifically within the Soft Actor-Critic (SAC) algorithm. The following is a broad assessment of its strengths across key dimensions.

2. Originality: The originality of the paper is moderate. The originality of this work is low. The paper simply combines two well-established techniques: the Soft Actor-Critic (SAC) algorithm and prototype-based explanation methods. Neither component is significantly modified or improved upon. The integration of these elements is straightforward and does not represent a novel algorithmic or theoretical contribution. The approach applies an existing interpretability paradigm to a new domain but fails to demonstrate substantive innovation in either the RL algorithm or the explanation framework itself.

3. Quality: The technical quality of the work is adequate but could be strengthened. The experimental validation, while demonstrating the method's basic functionality, is somewhat limited in scope, relying on a narrow set of environments. A more comprehensive evaluation, including comparisons to other explanation baselines and a deeper, more quantitative analysis of the prototype fidelity, would be necessary to robustly validate the framework's effectiveness and general applicability.

4. Clarity: The paper is generally clearly written, effectively motivating the need for explainability in RL and providing a coherent high-level overview of the ProtoSAC framework. However, the clarity is occasionally hindered, particularly by a central framework diagram that is not fully intuitive. The flow of how prototypes are generated and then utilized for explanation could be described with more precise, step-by-step detail to improve reader comprehension.

5. Significance: The significance of the work is limited. While applying prototype-based explanations to deep RL is a novel combination, the insights gained are not particularly surprising or impactful. We already have a strong understanding from other fields that prototypes can cluster behaviors and identify decision patterns. Applying this method to an RL agent merely confirms these known properties in a new context, without yielding any novel or counter-intuitive findings about the agent's learning process. The results are confirmatory rather than groundbreaking, making the overall contribution feel incremental.

**Weaknesses:**

1. The experimental environments used are overly limited and simplistic. The chosen benchmarks—Pendulum, MountainCar, InvertedPendulum, and LunarLander—are relatively simple and classic in contemporary deep reinforcement learning (DRL) research. These environments feature low-dimensional state vectors rather than high-dimensional pixel inputs, and the agent operates under clear dynamical models with intuitively understandable optimal policies.

2. ProtoSAC exhibits limited scalability and robustness. Even in these simple settings, ProtoSAC shows a slight performance degradation compared to the standard SAC baseline, as evidenced in Figure 2 and Table 2. The policies required in these environments are inherently straightforward—primarily involving swinging, balancing, or landing—and yet the model already relies on 30 to 60 prototypes. In more complex tasks that involve long-term planning, hierarchical decision-making, or hidden variables, a significantly larger number of prototypes would likely be needed to adequately cover the state space. This raises serious concerns regarding the method's scalability and general robustness.

3. ProtoSAC is narrowly built upon specific assumptions of the SAC algorithm. The method integrates prototypes only with SAC and cannot be directly extended to other actor-critic algorithms, such as deterministic policy-based methods like DDPG. Furthermore, it does not readily adapt to environments with non-continuous action spaces or those requiring alternative distribution representations—for instance, bounded continuous actions modeled using Beta distributions.

4. Insufficient analysis of experimental results. The interpretability claims of ProtoSAC are not convincingly supported by Figures 3 and 4 or their accompanying analysis. Although the captions state that “Each prototype is represented by its associated state and the action Gaussian distribution,” Figure 3 only visualizes the Gaussian distributions of actions (a, b, c, d), without illustrating the corresponding states associated with the four prototypes. This omission makes it unclear which specific states these prototypes represent. While the figures show how the model makes decisions—by blending prototypes—they fail to illustrate why those decisions are made, since the reader cannot see which representative states the agent considers similar to the current observation.

5. The absence of XAI/Explainable RL comparative experiments. As one of the main contributions of this article, the interpretability of this method also requires corresponding comparative experiments as support. However, although this article made a performance comparison with the explainable Shared-PW-Net, no comparison was made in terms of explainability.

**Questions:**

1. Expand the experimental validation to more complex environments. To rigorously assess the robustness and scalability of the proposed method, it is crucial to test it on benchmarks with high-dimensional state spaces, such as those requiring processing of image or video inputs. Demonstrating competitive performance and meaningful interpretability in these challenging settings would significantly strengthen the paper's contributions.

2. Conduct a more comprehensive evaluation of interpretability. The current qualitative analysis should be supplemented with a comparative study against other explainable DRL methods, particularly post-hoc approaches like ProtoX. A quantitative and/or human-study-based comparison would more convincingly demonstrate the advantages and unique value of the intrinsic interpretability provided by ProtoSAC.

3. Enhance the ablation studies. The paper would benefit from an additional ablation experiment that investigates the impact of the prototype update mechanism. By presenting results from a model variant where this update process is disabled, the authors could quantitatively validate its necessity for maintaining performance and prototype quality throughout training.

4. Revise the visualizations in Figures 3 and 4. To fully deliver on the promise of prototype-based explanation, these figures must be modified to visually show the actual states associated with each prototype. The current figures only display the action distributions, which makes it impossible for a reader to understand why a specific action is chosen. Illustrating the prototype states is fundamental for validating the "this state looks like that" reasoning.

5. I suggest that the author conduct a more in-depth analysis of the relationship among state, prototype and action distribution. In fact, it is very important to understand which observed variables in state/prototype affect the generation of actions.

6. The chart placement of this article needs further adjustment to make it easier to read.

---

> ### Author Response · Authors · 2025-12-02
>
> - **The experimental environments used are overly limited and simplistic. Expand the experimental validation to more complex environments**
>
>     We opted for these environments following what was used in literature. Our model does not have any constraint that hinders the usage of pixel inputs as the state representation is handled by a state encoder which can be implemented as any neural network. To address the reviewer's concerns, we introduce 4 additional more complex environments: HalfCheetah, Humanoid, Hopper and CarRacing. In 3 out of 4 of these environments, we report comparable or better performances with respect to the black-box model and it's distilled interpretable counterpart (SharedPWNet).
>
>
> | Environment       | SAC                   | Shared-PW-Net         | ProtoSAC              |
> |-------------------|------------------------|------------------------|------------------------|
> | Hopper-v5          |  3380.251 ± 1.980   |  3374.412 ± 3.049  |    **3386.222 ± 8.855**   |
> | HalfCheetah-v5    |    **11098.333 ± 150.083**     |    10369.168 ± 1248.026      |  9875.183 ±  77.872   |
> | Humanoid-v5      |    4574.069 ± 1358.091    |  496.586 ± 133.035      |     **4953.814 ± 3.268**   |
> | CarRacing-v3        |  232.968 ± 143.422   |  -32.528 ± 9.124  |    **369.047 ± 189.355**   |
>
>
> - **ProtoSAC shows a slight performance degradation compared to the standard SAC baseline**
>
>     We acknowledge that ProtoSAC sometimes might attain slightly lower performance than the black-box SAC baseline. However, it is well established in the literature that interpretable models could trade a small amount of performance for transparency compared to their black-box counterparts. Our goal is not to obtain a perfectly optimal controller, but rather to design an agent that learns directly via reinforcement learning, without relying on imitation learning or post-hoc explanation procedures, as in many existing approaches. In this setting, ProtoSAC achieves equal, better, or at least comparable performance to the black-box baseline while providing intrinsic interpretability.
>
> - **These environments are inherently straightforward—primarily involving swinging, balancing, or landing—and yet the model already relies on 30 to 60 prototypes.**
>
>     We agree that, compared to more complex continuous-control benchmarks, these environments are relatively straightforward in terms of task description (swinging, balancing, or landing). However, this does not imply that they can be solved with a “simple” or highly compact interpretable model. The underlying dynamics involve non-linear physics, derivatives, and control, which are challenging to capture in reinforcement learning with very small interpretable architectures.
>     In this work, we propose, to the best of our knowledge, the first self-interpretable model that learns these tasks without relying on imitation learning. Existing state-of-the-art interpretable approaches that do use imitation learning typically rely on a similar number of prototypes to ours. In addition, our method automatically adapts the effective number of prototypes: we added in the paper an ablation study where we progressively deactivate prototypes at different percentile thresholds of $\tau$, and we observe that removing low-$\tau$ prototypes does not significantly affect performance, thus proving its robustness, except in the extreme case where only one or two prototypes remain active.

---

> > ### Author Response · Authors · 2025-12-02
> >
> > - **ProtoSAC is narrowly built upon specific assumptions of the SAC algorithm**
> >
> >     ProtoSAC is indeed designed as a prototype-based extension of SAC, as explicitly stated in the title and abstract, and we do not claim it as a general prototype-based RL framework applicable to all actor–critic methods. The focus of the paper is to show that integrating prototypes into SAC can yield interpretable policies in continuous action spaces while maintaining competitive performance. Exploring how similar ideas could be adapted to other algorithms (e.g., deterministic policy-gradient methods such as DDPG) is an interesting direction, but it lies outside the scope of this work. We see this as a natural avenue for future research rather than a limitation of the current contribution’s goals.
> >
> > - **Insufficient analysis of experimental results and absence of XAI/Explainable RL comparative experiments.**
> >     We thank the reviewer for the comment. We provide an experiment measuring a fidelity score of ProtoSAC with respect to SharedPWNet. We measure fidelity as the deviation of the action when the most active prototype is shut down. Higher fidelity scores indicate higher faithfulness of prototypes, as their removal highly impacts the decision process. We observe that our method outperforms SharedPWNet in all the tested environments.
> >
> > | Environment      | Shared-PW-Net        | ProtoSAC              |
> > |------------------|-----------------------|------------------------|
> > | Hopper-v5        | 0.182 ± 0.174         | **0.534 ± 0.354**      |
> > | HalfCheetah-v5   | 0.402 ± 0.282         | **0.696 ± 0.528**      |
> > | Humanoid-v5      | 0.050 ± 0.044         | **0.084 ± 0.021**      |
> > | CarRacing-v3     | 0.188 ± 0.064         | **0.383 ± 0.525**      |
> >
> >
> > - **Investigates the impact of the prototype update mechanism**
> >     Empirical results on the HalfCheetah environment suggest a decay of about 25% in reward. This is due to the fact that the prototypes make the model more prone to less exploration and using the prototoype update allows to explore different solutions, thus incentivating exploration.
> >
> > - **More in-depth analysis of the relationship among state, prototype and action distribution. Figures 3 and 4 must be modified to visually show the actual states associated with each prototype**
> >     We are not sure whether fully understand the reviewer’s concern. In Figures 3 and 4, we report both the prototypes and the corresponding action distributions, rather than only the action distributions. Specifically, the first row in each figure shows the prototypes, and the second row shows the action distributions.
> >
> > - **The chart placement of this article needs further adjustment to make it easier to read**
> >     We have revised the placement of the figures/charts to improve readability and to ensure that each one appears closer to the corresponding discussion in the text. We hope that these adjustments make the article easier to follow

---

### Official Review · Reviewer_3FXf · 2025-10-31

**Soundness:** 3
**Presentation:** 3
**Contribution:** 3
**Rating:** 6
**Confidence:** 4

**Summary:**

The paper proposes an extension to current interpretable by design deep reinforcement learning algorithms by working prototypes into the training of soft actor critic. The method essential works by incorporating prototypes into the actor during training with the relevant gaussian processes. Comparisons are made against PW-Net variants and SAC vanilla algorithms, showing better performance than baseline interpretable by design methods, and almost equal performance to SAC.

**Strengths:**

The paper (as far as I know) is the first to propose a method to train interpretable by design deep RL agents from the ground up, which represents a significant contribution.

Appropriate baselines are chosen, and the environments tested sufficient I believe.

The ability to work in continuous action spaces is a big plus, as this represents most real-world deployment tasks of RL agents such as co-pilots etc.

**Weaknesses:**

I miss some qualitative analysis of the prototypes themselves, it is difficult to say how useful the final system would be for an end user, although to be fair this is a byproduct of most interpretability papers in the ML conferences.

The paper doesn’t consider pixel state spaces as far as I understand, which is a significant limitation. Would this mean the method is not applicable to deep learning problems? Or did you use pixel state spaces for these problems? If it is purely symbolic, I don't think it's fair to call this an interpretable by design deep RL algorithm, but I'm willing to hear a counter argument.

**Questions:**

I might have missed it in the manuscript, but did you use the symbolic or pixel state spaces for these problems?

Do you have any idea as to the qualitative properties of the prototypes? How easy would it be for a user to gain some kind of understanding of the model globally?

In the forward pass are you taking the state encoding’s similarity to all prototypes? I am wondering if the final output is a calculation traced back to all prototypes, which could be difficult to interpret.

---

> ### Author Response · Authors · 2025-12-02
>
> - **Qualitative analysis of the prototype**
>
>     In this paper, we do not focus on a user's perception of interpretability with respect to the agent. Instead, our emphasis is on developing mathematical constraints within the model to enable inspection of its decision-making process. While there is active debate in the literature regarding how users perceive explanations, we deliberately avoid this analysis, because user-based evaluation often reflects expectations rather than uncovering true factors influencing the model. Nevertheless, in response to reviewers' suggestions, we provide a quantitative analysis of explanation fidelity, measured as the action deviation when the most active prototype is shut down, demonstrating how key prototypes affect the agent’s decisions. This approach confirms that the prototypes offer faithful explanations aligned with the actual behavior of the model. Specifically, we compare the fidelity of our explanations to those produced by Shared-PWNet in the following table.
>
>
> | Environment      | Shared-PW-Net        | ProtoSAC              |
> |------------------|-----------------------|------------------------|
> | Hopper-v5        | 0.182 ± 0.174         | **0.534 ± 0.354**      |
> | HalfCheetah-v5   | 0.402 ± 0.282         | **0.696 ± 0.528**      |
> | Humanoid-v5      | 0.050 ± 0.044         | **0.084 ± 0.021**      |
> | CarRacing-v3     | 0.188 ± 0.064         | **0.383 ± 0.525**      |
>
>
>
> Additionally, regarding the question, "How easy would it be for a user to gain some kind of understanding of the model globally?", this reduces to identifying the most important prototypes and their associated Gaussian action distributions. Together, these elements provide a complete description of the model and enable a comprehensive understanding of its global behavior. More importantly, we observe, that thanks to our regularization losses, we force the model to use the smallest possible amount of prototypes, thus making explanations as simple as possible. We added in the paper where we show that the model retains strong performances even when least important prototypes (those associated with a small $\tau$) are removed.

---

> > ### Author Response · Authors · 2025-12-02
> >
> > - **Pixel States**
> >
> > We would like to emphasize that ProtoSAC can take all types of input including 1-d vectors, images, or even sequences and graphs. Indeed prototype-based models were first developed with images and the image application would be straightforward. This is due to the fact that the state/prototype representation is handled by the state encoder $f_s$ which can be implemented as any neural network (MLP, CNN, ...).
> >  We evaluate this hypothesis in the CarRacing-v3 environment by training our model with a CNN encoder. We compare the performance with the baseline trained using the same hyperparameters, as well as with the Shared-PW-Net trained using the SAC baseline. As shown in the table below, ProtoSAC clearly outperforms Shared-PW-Net, which is unable to solve the environment.
> >
> >
> >  | Environment       | SAC                   | Shared-PW-Net         | ProtoSAC              |
> > |-------------------|------------------------|------------------------|------------------------|
> > | CarRacing-v3        |  232.968 ± 143.422   |  -32.528 ± 9.124  |    **369.047 ± 189.355**   |

---

> > > ### Author Response · Authors · 2025-12-02
> > >
> > > - **The forward pass uses similarity to all prototypes?**
> > >
> > > Yes, in the forward pass, we calculate the similarity with all the prototypes, but weighted by the importance weight $\tau$, in this way, the most used prototypes have more influence than others. Thus, to interpret the action, one does not necessarely have to watch all the prototypes but only the most similar among the important ones.

---

### Author Response · Authors · 2025-12-02

We thank the Reviewers and Area Chairs for their thoughtful feedback. Below is a summary of our responses to the primary points raised. In this work, we introduce ProtoSAC, a new framework for training self-explainable reinforcement learning agents in continuous action-space environments. The reviewers raised the following main concerns:

- **Lack of complex environments and image states:** We expanded our evaluation to include four additional challenging environments (HalfCheetah, Humanoid, Hopper) including a visual (image-based) setting with CarRacing. These tests corroborate our main findings: ProtoSAC achieves performance comparable to both black-box SAC and state-of-the-art self-explainable agents that rely on distillation, even in more complex domains.

- **Explanation evaluation:** Rather than conducting a user study, which may introduce user expectation bias, we implemented a fidelity metric to quantify how prototypes contribute to action selection. This ensures explanations accurately reflect underlying model behavior. We further demonstrate that ProtoSAC can produce highly concise explanations by leveraging only the prototypes necessary for each decision.

- **Prototype count and interpretability:** Some reviewers noted that using as many as 60 prototypes could impede interpretability. Our results show that with our proposed regularization techniques, ProtoSAC often learns to use markedly fewer prototypes, even as few as 5 in simpler environments, supporting greater model transparency.

Other points raised by reviewers have been answered in detail in the response section. We believe our revisions address the reviewers' concerns, and we respectfully request consideration of these clarifications by the Area Chairs.

---

### Meta-Review · Area_Chair_L1tU · 2026-01-12

**Summary:**

The reviewers raised many important concerns and unfortunately the authors were unable to respond to the concerns before the discussion freeze. Among the primary concerns identified by the reviewers, i.e. lack of complex environments, explanation evaluation, and prototype count and interpretability, only the weakness regarding environments has been meaningfully addressed in the rebuttal (with additional experiments). Regarding explanation evaluation, the authors assert that a quantitative analysis of the prototype fidelity metric, possibly via a user study, is unnecessary. However, the reviewers do not seem to have found this argument compelling as presented in the paper. The prototype count aspect has not been sufficiently addressed in the rebuttal, as the key concern from the reviewers is that while for simpler environments the method chooses a small number of prototypes, this framework would fail to provide interpretability for complex environments which would require a large number of prototypes. From the author response, it seems this is a limitation that needs to be explicitly acknowledged in the paper, or refuted with experiments on significantly more complicated environments than those added in the rebuttal (HalfCheetah, Humanoid, Hopper).

**Reviewer Concerns:**

As mentioned above, lack of complex environments, explanation evaluation, and prototype count and interpretability were the primary concerns raised by the reviewers. The rebuttal addresses the concern about complex environments with additional experiments on three new environments, HalfCheetah, Humanoid, and Hopper. The concerns about explanation evaluation, and prototype count and interpretability are not addressed sufficiently.

**Reviewer Scores:**

None of the reviewers are likely to have changed their scores.

---

### Decision · Program_Chairs · 2026-01-26

Reject